# The Impact of COVID-19 on the Sustainable Development Goals: Achievements and Expectations

**DOI:** 10.3390/ijerph192316266

**Published:** 2022-12-05

**Authors:** Cathaysa Martín-Blanco, Montserrat Zamorano, Carmen Lizárraga, Valentin Molina-Moreno

**Affiliations:** 1Department Business Organization I, University of Granada, 18011 Granada, Spain; 2Department of Civil Engineering, University of Granada, 18011 Granada, Spain; 3Department of Applied Economics, University of Granada, 18011 Granada, Spain

**Keywords:** SDG, COVID-19, circular economy, bibliometric analysis, systematic review

## Abstract

The COVID-19 pandemic has had a significant impact on almost all the Sustainable Development Goals (SDGs), leaving no country unaffected. It has caused a shift in political agendas, but also in lines of research. At the same time, the world is trying to make the transition to a more sustainable economic model. The research objectives of this paper are to explore the impact of COVID-19 on the fulfilment of the SDGs with regard to the research of the scientific community, and to analyze the presence of the Circular Economy (CE) in the literature. To this end, this research applies bibliometric analysis and a systematic review of the literature, using VOSviewer for data visualization. Five clusters were detected and grouped according to the three dimensions of sustainability. The extent of the effects of the health, economic and social crisis resulting from the pandemic, in addition to the climate crisis, is still uncertain, but it seems clear that the main issues are inefficient waste management, supply chain issues, adaptation to online education and energy concerns. The CE has been part of the solution to this crisis, and it is seen as an ideal model to be promoted based on the opportunities detected.

## 1. Introduction

Sustainable development is not a recent or novel concept in the 21st century, although it is acquiring special relevance as a consequence of the externalities of the traditional model. In fact, the concept emerged in the early 1970s as a move to protect the environment and ensure development without the associated destruction [1], and was defined by the World Commission on Environment and Development (WCED) in 1987 as “meeting the needs of the present without compromising the ability of future generations to meet their needs” [2]. There have been many supranational strategies to promote sustainable development, from Agenda 21 in 1982 [3], the Millennium Development Goals at the beginning of the 21st century [4], and the recent Sustainable Development Goals (SDGs) established in 2015, including several evaluation milestones [5]. Agreed on by 193 countries, the SDGs are operationalized through 169 targets and 213 measurable indicators that form a global action plan [6]. They aim to address the systemic barriers to sustainable development in three dimensions—social, economic and environmental—with universal application under the premise of a growing interconnected world [7]. The SDGs are classified into five groups, named the “Five Ps”: (1) People (SDG 1, no poverty; SDG 2, zero hunger; SDG 3, good health and well-being; SDG 4, quality education; SDG 5, gender equality); (2) Planet (SDG 6, drinking water and sanitation; SDG 12, responsible consumption and production; SDG 13, climate action; SDG 14, underwater life; SDG 15, life on earth); (3) Prosperity (SDG 7, clean and affordable energy; SDG 8, decent work and economic growth; SDG 9, industry, innovation and infrastructure; SDG 10, reduction of inequality; SDG 11, sustainable cities and communities), (4) Peace (SDG16, strong institutions for peace and justice) and (5) Partnership (SDG17, partnerships to achieve the goals).

The social dimension of sustainability is addressed through the “People” goals. There is no consensus on its definition due to the divergence of approaches for the study of this aspect [8,9], and indeed, little academic attention has been focused on this dimension [10]. Its conceptualization faces other problems due to the inclusion of soft terms such as social capital [8], which causes further difficulties in its analysis. Landorf [11] proposed that this dimension is a binomial between social equity and community sustainability, which represent the most common terms used [8]. The most recent studies take an integrated approach due to the interconnected nature of the three dimensions [12].

The “Planet” goals focus on the environmental dimension of sustainability, understanding it as a natural science concept that obeys biophysical laws, seeking the “unimpaired maintenance of human life-support systems-environmental sink and source capacities” [13]. This concept is related to the resource-limited ecological economic framework of “limits to growth” [14].

The economic dimension fits under the “prosperity” goals. For a long time, economic policies were only applied to the distribution and allocation of resources, without paying attention to the scale of extraction from nature [15]. The 17 SDGs address challenges and take actions that can be grouped under the three sub-goals of ecological economics, which aim to move towards an efficient, just and sustainable economy [7]. This shift is related to efforts to embed sustainable finance in both private and public organizations, as well as policy initiatives to encourage responsible business conduct for sustainable development [16,17,18].

Finally, the remaining two Ps (Peace and Partnership) work as facilitators for the rest of the dimensions [5]. Peace is related to SDG 16, which is focused on improving democracies and protecting human rights, whereas Partnership is associated with SDG 17, whose primary aim is to forge alliances between public and private entities in order to achieve international cooperation and to cope with global issues such as climate change or economic crises.

Nowadays, the traditional linear economic model of “take, make and throw” has become unsustainable [19,20], resulting in the need to transition to more sustainable socio-technical systems [21,22]. The externalities of the linear production model are threatening the economic and environmental sustainability of our planet, thus causing natural ecosystems to be in jeopardy [23,24,25,26,27,28]. Similarly, society faces high rates of unemployment, and poor working conditions, leading to social vulnerability, conceptualized through poverty and increasing inequalities [29,30]. Sustainability requires the development of a balanced production system, taking into consideration economic, social, environmental and technological aspects [31], with the Circular Economy (CE) being a new paradigm that contributes to the positive reconciliation of all these elements [23,32].

In this context, CE is defined as an industrial economy that is restorative and regenerative by concept, intention and design [23,33,34]. It brings together diverse schools of thought [35], including biomimetics [36], performance economics [37], natural capitalism [38], regenerative design [39], cradle-to-cradle [40], blue economy [41] and industrial ecology [42]. Moreover, it is considered by the new circular economy action plan (CEAP) adopted by the EU in 2020 as one of the main building blocks of the European Green Deal, Europe’s new agenda for sustainable growth, and it is a prerequisite to achieve the climate neutrality target and to halt biodiversity loss. In fact, the transition to a CE will reduce pressure on natural resources and create sustainable growth and jobs. The field of knowledge of ecological economics or the green economy is unarguably at the roots of the CE, interwoven in its three dimensions of sustainable action [43,44]. Therefore, actions in the CE are closely related to the achievement of the SDGs [45], in addition to sharing a bias towards degrowth and green growth research that seeks efficient allocation [33]. Thus, five SDGs have strong synergies and a direct relation with CE practices, concretely, SDG 6 (clean water and sanitation), SDG 7 (affordable and clean energy), SDG 8 (decent work and economic growth), SDG 12 (sustainable consumption and production), and SDG 15 (life on land). Meanwhile, SDG 1 (no poverty), SDG 2 (zero hunger) and SDG 14 (life below water) are impacted by CE practices for the most part indirectly. In contrast, SDG 4 (quality education), SDG 9 (industry, innovation and infrastructure), SDG 10 (reduced inequalities), SDG 13 (climate action), SDG 16 (peace, justice and strong institutions) and SDG 17 (partnerships for the goals) show a potential relationship with CE practice that could maximize their progress [45]. Figure 1 shows the relationships between the Five Ps (first circle), the SDGs (second circle), the CE (third circle) and the ecological economics goals (the three discs orbiting the larger one).

According to Sachs et al. [46], during the period 2015–2019, the world made progress towards the SDGs at a rate of 0.5 points per year, which is not fast enough to meet the 2030 deadline. Figure 2 shows the evolution of the SDG Index score since 2010.

By 2022, the 2030 Agenda is halfway to its target date and the COVID-19 pandemic has not favored the achievement of its goals. Even if the pandemic has not completely wiped out the progress made to date on some of the targets, it has certainly has made it more challenging, with progress inching along at a rate of 0.1 points per year [46]. However, even in a world with no COVID-19, the global targets would not be met [47,48].

As declared by the World Health Organization on 11 March 2020, the global pandemic has been a historically unprecedented episode in terms of financial investment in research that has prompted the generation of scientific literature in all fields on sustainability, especially on the impact of the pandemic and the forecasting of possible futures [47]. Even though the impact of the pandemic on SDGs is still uncertain [49,50], the measures taken by governments to contain the virus spread have shattered the basis of a globalized world that relied on international trade, and forced states to compete for scarce resources. The lockdown brought about the shutdown of the economy, putting the SDGs and the transition to the CE at the center of the debate [49,51,52].

The research objectives of this paper are to explore the impact of COVID-19 on the fulfilment of the SDGs with regard to the research of the scientific community, and to analyze the presence of the circular economy (CE) in that literature. To this end, this research applies bibliometric analysis and a review of the literature, and answers the following questions:Q1. What are the main characteristics of this line of research?Q2. What are the main thematic areas and the most relevant publications that address the impact of COVID-19 on the SDGs?Q3. Who are the most productive authors, institutions, countries and journals?Q4. What are the main international cooperation networks?Q5. What are the main current trends in research on COVID-19 and SDGs?Q6. What have been the main contributions of the CE to the SDGs during COVID-19?

In general, studies that apply a systematic review of the literature are valuable to understanding the leading edge research in a field, but an additional analysis of the literature using bibliometric methods can provide results that have not been detected in the other reviews [53]. Thus, bibliometric research, analyzing large volumes of scientific data, has an increasing scientific value in responding to the current public health emergency of international concern [54]. In this case, the pandemic has generated enormous attention in the scientific community, reflected in the large volume of articles published during the last two years, but it has been shown that COVID-19-related reviews have been limited and fragmented in particular areas [12]. This paper fills this gap, applying bibliometric analysis to explore the implication of the impact of COVID-19 on the fulfillment of the SDGs and the presence of the circular economy (CE) in the literature.

## 2. Materials and Methods

### 2.1. Bibliometric Analysis

Scientometric or bibliometric analysis is a research methodology whose main objective is to identify, organize and analyze metadata to examine the evolution of an area of knowledge during a specific period of time [55,56,57,58]. The systematic literature review can provide a state of the art and identify gaps and potential areas for future research in the literature, but the procedure followed must be replicable [59].

### 2.2. Methodological Procedure

To this end, this study works under the SPAR-4-SLR protocol established by Paul et al. [59], which consists of three stages and six sub-stages. Table 1 summarizes how the protocol has been applied in this research.

The search took place in March 2022 and the study was conducted in three phases. First, search criteria were selected to identify records in the repository (identification phase). Then, having obtained the records that met the search requirements, the data were exported for analysis using VOSviewer v. 1.6.18 software (analysis and visualization phase). Finally, connections and associations between the scientific documents were established and a discussion (results and discussion phase) took place.

#### 2.2.1. First Phase: Identification

The Scopus scientific database was used for the data search, although the main scientific repositories such as Web Of Science, PubMed and Google Scholar were consulted, following the recommendations of Harzing and Alakangas [60] and Mongeon and Paul-Hus [61]. The reasons for using Scopus were as follows. (a) It is the repository with the largest volume of information on authors, countries and institutions [62,63]; (b) it has the highest volume of articles meeting the scientific quality requirements for peer review [62], [64]; and (c) the coverage it provides compared to Web of Science and other repositories is greater, while its metrics are highly correlated [65,66]. Consequently, it has been selected by Baas et al. [67] and Donthu et al. [68] as the most suitable repository to apply this research methodology.

The search in the Scopus repository was carried out using the fields “Article title, Abstract, and Keywords”, with the following search terms selected: TITLE-ABS-KEY (“COVID-19” OR “coronavirus disease 2019” OR “SARS-CoV-2” OR “coronavirus” OR “coronavirus” OR “coronavirus infection”) AND TITLE-ABS-KEY (“sustainable development goals” OR “sustainable development” OR “sdgs” OR “sdg” OR “Agenda 2030”). As a result, a total of 2453 documents were obtained that met the search criteria.

Next, exclusion criteria were applied. Firstly, following the recommendations of Paul et al. [59], research articles only were selected, as these are published on the basis of scientific novelty and satisfy the scientific quality criterion of peer review. Consequently, the number of papers that met the search criteria was reduced to 1483. Then, a time horizon restriction to the years 2020 and 2021 was applied, as the first notification of the existence of a cluster of cases of the virus in Wuhan was made on 31 December 2019. Therefore, all previously published articles are not in line with our research objective, again reducing the number of documents meeting the search criteria to 1148. Finally, the language criterion was applied, selecting only research articles published in English, which reduced the total number of documents to 1093, which are those that make up the sample of our bibliometric and systematic review.

Therefore, the final search string was as follows: (TITLE-ABS-KEY (“COVID-19” OR “coronavirus disease 2019” OR “SARS-CoV-2” OR “coronavirus” OR “coronavirus” OR “coronavirus infection”) AND TITLE-ABS-KEY (“sustainable development goals” OR “sustainable development” OR “sdgs” OR “sdg” OR “Agenda 2030”)) AND (LIMIT-TO [PUBYEAR, 2021) OR LIMIT-TO (PUBYEAR, 2020)) AND (LIMIT-TO (DOCTYPE, “ar”)) AND (LIMIT-TO (LANGUAGE, “English”)).

#### 2.2.2. Second Phase: Analysis and Visualization

For the analysis and visualization of the documents, VOSViewer v.1.6.18 was used, which allows clustering and word processing [69,70]. The tool is useful for clustering networks of a large number of documents, keywords, authors or institutions [71].

Accordingly, based on the documents that met the search criteria, the cooperation networks between authors, institutions and countries through the co-citation method were analyzed. These international networks provide insight into the relationships between researchers and the dissemination of knowledge [72], while favoring the design of new research by generating synergies that contribute to the exchange of ideas [73].

An analysis of the clustering of keywords contained in the documents on COVID-19 and SDGs using the co-occurrence method was also conducted, as these are considered representative of their content [74]. Co-occurrence is based on the fact that records sharing the same keywords are similar [75,76]. This allows us to analyze the evolution of the topics covered in research papers [77], creating a picture of the line of research [78].

#### 2.2.3. Third Phase: Results and Discussion Phase

Finally, the third phase entailed analyzing authors, institutions, countries, journals and international cooperation networks, as well as keywords, to identify research trends on the impact of COVID-19 on the SDGs. Together with the systematic literature review, this contributed to resolving the research questions posed and presenting the discussion and conclusions of this research work.

## 3. Results and Discussion

### 3.1. Evolution of Scientific Production

This section presents the results on the main characteristics of the scientific production of the impact of COVID-19 on the SDGs in the period 2020–2021 (Table 2). The total number of publications during the two years is 1093 articles, 77% published in the last year. The growing interest in the subject is shown by the 233% increase in the number of authors, 49% in the case of countries, 235% in the case of institutions and 168% in the number of journals.

Regarding the number of citations, the articles published in 2020 received 431 citations during that first year, which is an unusually high number, highlighting the great interest in the impact of the pandemic on the SDGs. Moreover, publications in the second year received ten times more citations than those in the first year, showing a very high productivity. In contrast, the average number of authors per publication did not change substantially because of the simultaneous increase in the number of authors and publications.

### 3.2. Most Influential Subject Areas and Publications

This section pertains to the second question regarding the most productive areas of knowledge. Since an article can be in more than one area [79], Figure 3 shows a large number of articles, which are classified into 27 thematic areas, with 83.75% accumulated in seven areas. The Social Sciences area is the most productive, with 565 articles representing 23.91% of the total scientific literature. This is followed by Environmental Sciences (*n* = 484, 20.48%), Energy (*n* = 348, 14.73%), Business, Management and Accounting (*n* = 162, 6.86%), Medicine (*n* = 153, 6.47%), Engineering (*n* = 145, 6.14%) and, finally, Economics, Econometrics and Finance (*n* = 122, 5.16%). The presence of these areas is evidence of the impact of the pandemic on the three dimensions of sustainability, as it is a phenomenon studied from an economic, social and environmental perspective, as well as the interconnection between the SDGs affected. The areas involved in this subject are similar to those involved in studying the CE, as Belmonte-Ureña et al. [80] pointed out, with the nuance that in our subject, the Social Sciences have been predominant alongside the Environmental Sciences, while the more technical areas have experienced a greater focus in the subject of the CE.

Table 3 shows the most relevant publications in this research area according to the total number of citations.

At the beginning of the pandemic, Zambrano-Monserrate et al. [81] revealed the indirect effects of COVID-19 on the environmental dimension of sustainability in two ways. On the one hand, the positive effects include the improvement of air quality and the reduction of noise and environmental pollution due to the drastic reduction in activity due to the protection measures. At the same time, however, these measures produced negative effects such as an increase in waste generation due to the growing consumption of single-use plastics, as well as a decrease in recycling and waste management. They concluded by indicating that, given the recovery of economic activity, the temporary positive effects will not be enough to offset the negative effects of pollution. In fact, Coccia [82] highlighted the threat that contamination represents for the challenge of improving resilience to future pandemics. In his study of 55 Italian provincial capitals, he concluded that persistent air pollution was the determining factor for virus transmission, rather than the effect of direct person-to-person contact. Pirouz et al. [84] also addressed the prevention of future Coronavirus epidemics through the development of a predictive model for the occurrence of positive cases. Unlike Coccia [82], they studied only one city and concluded that the determining factors were humidity and temperature, that is, the relative humidity in the main case study, with an average of 77.9%, positively affected confirmed cases, and maximum daily temperature, with an average of 15.4 °C, negatively affected confirmed cases. Additionally, this article was the first to include predictive techniques based on massive data, being part of the advance of the fourth industrial revolution. Meanwhile, Vanapalli et al. [83] focused on the problem of plastic pollution, recommending measures to improve the management of this waste, moving towards a model that uses environmentally friendly materials and increasing investment in sustainable technologies that allow progress in the transition to a circular model to fight against future pandemics. Finally, in the environmental dimension, Ibn-Mohammed et al. [49] highlighted the need to make a transition from the linear to the circular model, considering that we are facing an opportunity to promote a low-carbon economic model in a more resilient world; for this, they provide recommendations according to key sectors.

For their part, Ilyas et al. [85] considered the social dimension of sustainability, emphasizing the environmental and health risks posed by mismanagement of biomedical waste (COVID-waste). They carried out an analysis of disinfection techniques with the intention of providing information for the prevention of future pandemics, thus also focusing on improving resilience. The study by Yeasmin et al. [86] highlights one of the main effects of the pandemic: the worsening of mental health, especially in children, as a consequence of confinement. They also provided a series of recommendations to achieve SDG 3, such as the implementation of psychological intervention strategies and the improvement of the sociodemographic conditions of households, including economic security, education, childcare and job stability, all which have been severely affected by this crisis. Along the same lines, Leal Filho et al. [87] warned of other diseases that could occur, as well as the negative impact on mental health as a result of protection measures against the virus, concluding that the pandemic is a serious threat to the achievement of the SDGs due to its severe impact in all areas, and they press for greater action to accomplish the SDGs.

Finally, Amankwah-Amoah [88] centered the economic dimension of sustainability, specifically the performance of airlines in relation to their Green Business Practices (GBP). His analysis shows that some companies evaded their environmental commitments by prioritizing market survival and cost reduction. Galvani et al. [89] also discuss airlines and tourism, it is surprisingly the only article with a positive view on the effect of the pandemic, specifically on the change of humanity towards a mindset aligned with the SDGs. This coincides with the initial speculation about the positive impacts of the pandemic, but which two years later seem far from reality.

For the analysis of the impact of COVID-19 on the SDGs, it has been considered appropriate to include a column indicating the SDGs addressed by the ten most cited articles, using the new functionality of Scopus. Elsevier data science teams have built extensive keyword queries, supplemented with machine learning, to map documents to SDGs with very high precision.

Thus, six out of ten articles discuss SDG 3—good health and well-being, which is the only “People” goal studied among the ten most cited articles. Hence, goals linked to People have received insufficient attention from the science community regarding COVID-19 impacts on SDGs, despite the fact that it is a health issue. The “Planet” goals and the environmental dimension are addressed through three goals. Concretely, three articles study SDG 13—climate action, one studies SDG 14—life below water, and five study SDG 12—responsible consumption and production. Finally, in relation to the articles examining the “Prosperity” goals and the economical dimension, there are three studies that focus on SDG 8—decent work and economic growth, four on SDG 9—industry, innovation and infrastructure and only one deals with SDG 10—reduced inequalities. These results indicate that, in the first stages of the pandemic, the economic consequences of and solutions to the pandemic were the issues that received the most attention from the scientific community (measured through the number of citations received per article). In addition, these goals bear upon the CE paradigm, especially SDGs 12, 8 and 9.

### 3.3. Authors’, Journals’, Institutions’ and Countries’ Productivity

This section presents the productivity results of authors, institutions, countries, journals and their global cooperation networks.

Table 4 shows the ten most productive authors on the topic of the impact of the pandemic on the SDGs and their main characteristics.

Among the ten most productive authors, seven come from Asia and one of these, Dang, T. T., leads the list, with twice as many citations as the second and third most productive authors, despite having only been published in the last year. Meanwhile, Leal Filho, W., the only European author, is the second most productive author, with the same number of articles as Dang but with only half as many citations, even though published during the two years studied. Shaw, R. and Adelodun, B. also published both years, which is not unexpected, taking into account the novelty of the subject. However, the latter two authors are the most cited among those studying the impact of the pandemic on the SDGs and have the best ratio of total number of citations to total number of published articles. Ali, S. M. focuses his studies on the impact of the pandemic on supply chains (SC) and its implications for the SDGs; specifically, he explores the drivers for improving the sustainability of SCs [90] and the challenges of maintaining the vaccine SC [91] and the humanitarian SC [92], all with a decision-making approach and in relation to a wide range of SDGs such as 8, 10, 12 and 3. Meanwhile, Allam, Z. explores the future of post-pandemic cities in terms of their socio-economic sustainability through the paradigm of “the 15-minute city” [93], smart cities through 6G [94] and the achievement of inclusive cities [95], all related to SDG 11.

Figure 4 shows the cooperation network based on the co-authorship analysis. The criteria used for clustering were: applying the fractional counting method, ignoring documents with more than 25 authors, selecting an interaction of at least two co-authored published research papers and the association strength method for normalization [69].

International cooperation is generally weak. Leal Filho, W. is the only author from the ten most productive authors to appear in the cluster. Through his five documents, he establishes the connection between the cluster and an international cooperative network. Three authors are from Brazilian institutions (Quelas, O. L. G. from Universidade Federal Fluminense, Anholon, R. from Universidade Estadual de Campinas, Fritzen, B. from Universidade de Passo Fundo and Salvia, A. L. from University of Passo Fundo), Rampasso, I. S. is from Universidad Católica del Norte (Chile), Wall, T. is from Liverpool John Moores University (United Kingdom) and Doni, F. is from University of Milano-Bicocca (Italy).

Table 5 shows the ten most productive institutions. Chinese Academy of Sciences leads the ranking with 15 articles and an H index of 7, which corresponds with the high number of Chinese authors on this topic. However, with half as many articles as the former, UNSW Sydney and Texas A&M University have the best ratios of citation per articles (15 and 14.71, respectively). Moreover, the latter has a 100% cooperation index, followed by Organisation Mondiale de la Santé, University College London, UNSW Sydney and London School of Hygiene & Tropical Medicine.

Figure 5 shows the network of cooperation between organizations based on the co-authorship of articles. Initially, 2969 organizations were detected, but choosing those with a minimum of two documents resulted in just five organizations being connected, meaning that the network is not solid, due to of the marked lack of cooperation. 

In relation to the quality of academic institutions, the most productive countries (Table 6) are China, the United States and the United Kingdom. Spain, the fourth, receives fewer citations than Italy despite being more productive. In general, the cooperation index for countries is low except for Australia. This may indicate that the topic has been studied among researchers from the same country, but from different institutions and especially from the countries most affected by the pandemic such as China, the United States, the United Kingdom and Italy.

Figure 6 shows the international collaboration networks of countries based on the co-authorship of articles. The colors show the networks and the size of the circles indicates the productivity of the networks based on the number of documents The limit was set at an interaction of at least 10 studies that were published with international co-authorship, reducing the number of countries from 141 to 47. They were grouped into six clusters. The first cluster, colored in red, is composed of 14 countries, led by Italy, Spain and Germany. The green cluster is made up of 10 countries and is led by India and South Korea. The blue cluster includes eight countries, led by Australia, and shows an interrelation between Pacific countries. The yellow cluster is made up of seven countries and is led by China, which is the country with the most publications. The purple cluster is composed of six countries and is led by the second and third most productive countries, the United Kingdom and the United States. Only Iran and Turkey are included in the light blue cluster. Clusters, most notably the purple cluster, show that connections are more frequent if the organizations are from the same continent.

Table 7 shows the ten most productive journals addressing the impact of the pandemic on the SDGs, which account for 18% of the articles. According to SCImago Journal Rank, all of them belong to the first quartile and chiefly relate to the main subject areas such as energy, environmental science or the social sciences. The most prolific journal is *Sustainability*, which has 62% of the articles published in the top ten most productive journals and the best H-index of the articles in that area of research. Nevertheless, the journal *Science Of The Total Environment*, seventh in the ranking, has the highest total citations and the second best H-index after the journal *Renewable And Sustainable Energy Reviews*, which ranks last in terms of number of articles, but has the best quality indexes, as it is the journal with the greatest influence (3.68 impact factor). Finally, four journals are Swiss, three of which are leading the ranking, four are Dutch, two are British and one is American.

### 3.4. Which Have Been the Most Frequently Undertaken Problems and Results? Which SDGs Have Received the Most Attention?

Co-occurrence analysis was applied using the indexed words as the unit and the fractional counting method. A thesaurus file was also introduced to eliminate search words and countries, as well as to standardize words appearing in singular and plural. A limit of at least 10 occurrences was then set, which reduced the number of terms from 4291 to 134. Five clusters emerged (Figure 7), each one representing a theme, and they were ordered according to the number of documents included (for example, Cluster 1 is the first because it contains the largest number of documents). This section contains the most frequently undertaken problems and the most frequently obtained results by cluster; they are grouped according to the three dimensions of sustainability. The social dimension includes Cluster 1, focused on health, which is logical given that it is the main issue caused by a pandemic. Cluster 5 is focused on education, the economic dimension includes Cluster 2 and the environmental dimension includes the third and fourth clusters, as these refer to energy, waste management and pollution.

#### 3.4.1. Social Dimension

##### Cluster 1: Public Health Dimension

Colored in red, this cluster contains 42 elements. It focuses primarily on SDG 3 because public health is the central theme and it involves issues related to infection, prevention and health policy. In addition, it is connected with the consequences of lockdown and the socio-economic implications, especially in developing countries.

The key point is to develop policies that focus on prevention and build resilience to disruptive events such as the pandemic, which are expected to increase in frequency. Ilyas et al. [85] assessed the management of bio-medical waste products, considering them as health and environmental risks, for which they advocate the use of disinfection techniques. Yeasmin et al. [86], meanwhile, highlighted the impact of the pandemic on mental health, a line of research in psychology that draws attention to the effect of isolation on young people and children and its relationship with socioeconomic conditions. In addition, rising youth unemployment is increasing psychological stress [96].

Meanwhile, other disease prevention programs have seen an overall reduction in the attention they receive in developing countries [87]. For these reasons, Adelodun et al. [97] proposed monitoring virus transmission through wastewater analysis as a sustainable preventive measure, reducing the cost for these countries. On the other hand, Visconti and Morea [98] recommend the promotion of digital technologies in healthcare through public–private partnerships to reduce costs, decongest hospitals and improve disease surveillance.

##### Cluster 5: Educational Dimension

Cluster 5 in light blue contains 11 elements. It focuses on education, especially at the university level, and the introduction of online teaching with the implications for the teacher and the student. Thus, it is mainly linked to SDG 4, but also to SDGs 9 and 12.

Twenty-first century education has long included information and communication technologies in its procedures, but, in the wake of the pandemic, the use of online learning platforms has intensified [99]. This situation poses serious problems for access to education in developing countries and for students in rural areas [100], for whom measures are needed to address barriers such as poor access to electricity and the relevant equipment. In contrast to Wang and Huang [54], who also studied the impact of COVID-19 on SDGs, our research found that developed countries were not primarily focused on SDG 4. There are many studies from Asia, South America and Africa. For example, Rodriguez-Segura et al. [101] from Mexico and Edelhauser and Lupu-Dima [99] from Romania studied emergency remote teaching, finding that their countries were unprepared for the switch to e-learning, but that the pandemic has been a turning point in their transition. It represents an opportunity, especially for higher education, to become more sustainable and accessible, although it requires a change in the organizational culture [100,102].

Studies carried out on this topic have also considered the digital skills of educators. Portillo et al. [103] found that there is a digital divide among teachers according to gender, age and educational level; what is especially concerning is that they noticed lower technological competence at the lower levels of education, which are more vulnerable to deficiencies in remote teaching. Meanwhile, Tran et al. [104], investigated how to elevate students’ learning habits to achieve quality education and its relation to socioeconomic conditions.

#### 3.4.2. Economic Dimension

##### Cluster 2: Economic Dimension

Colored in green, this cluster contains 32 elements. Its focus is on the economic dimension of the crisis and, in particular, on tourism. Adopting a strategic approach is a key point in this cluster, as well as the necessity to innovate and use knowledge for new solutions. It is linked to SDGs 8, 9, 11 and 12.

One line of research focuses on the economic consequences of the pandemic and the need to switch to a more sustainable and resilient economic system. For example, the shutdown of air traffic during the pandemic dealt a severe economic blow to countries heavily dependent on tourism, but current air travel is a highly polluting activity that needs to be replaced by a smaller, less economically vulnerable alternative models that take into account negative externalities [105]. In general, studies are exploring and calling for measures from the tourism sector to become more transformational and transcendent in order to achieve the SDGs [106]. From a much more positive outlook, Galvani et al. [89] considered that travel and tourism now have the opportunity to offer valuable experiences and to become a means to expand the global consciousness that emerged in the aftermath of COVID-19.

Cities must be rethought using prospective technologies in order to effectively manage this and future pandemics [107]. However, policymakers must proceed cautiously when introducing new technology; for example, the proposal by Shorfuzzaman et al. [108] to apply mass video surveillance to control the spread of the virus violates rights of privacy.

The pandemic has also affected commerce and environmental awareness, forcing businesses to implement e-commerce platforms and to orient themselves to the increasing profile of responsible consumers who are informed about sustainable production and consumption [109]. Similarly, Tchetchik et al. [110] pointed out that COVID-19 has driven a change in the behavior of consumers towards greater environmental awareness, but mainly as a consequence of threat and endeavors to cope with it.

Another discussion connected with this dimension is the value of care work. Authors such as Bahn et al. [111] call for the incorporation of lessons from feminist economics into the economic system, giving a proper place to the care work that was essential during the pandemic and, in general, to the achievement of human wellbeing.

Supply chains have received massive attention due to their far-reaching implications. In developing countries, where the informal economy is predominant, economic shutdowns and vulnerable supply chains have led to increased food insecurity [112], especially regarding supply chains for perishable foods [113]. Equally important are problems in the supply chain for vaccines, which are essential to curb infections [89].

#### 3.4.3. Environmental Dimension

##### Cluster 3: Energy Dimension

Cluster 3 in blue contains 29 elements. It is focused on renewable energy to combat climate change and energy dependence, and the need to invest in and streamline energy policy from a global perspective is highlighted. This cluster contains the term “Circular Economy”, but this will be analyzed in a separate section due to research question six. This cluster is linked to SDGs 7, 8, 9, 12 and 13.

Madurai Elevasaran et al. [114] tracked the impact of sustainable energy on the rest of the SDGs, demonstrating that energy transition is essential to cope with the new challenges. The energy sector has been under huge strains during the pandemic, which has driven investment in renewable energy [115]; the authors call on governments to develop short- and long-term strategies for clean energy efficiency, creating a win–win solution for economic recovery and energy supply chains [116]. It is also necessary to promote research into energy storage systems and technologies that reduce energy consumption and to facilitate entrepreneurship in the sector and the creation of energy communities [114]. Moreover, studies on the relationship between the energy, water and food supply sectors are needed in order to ensure resource security [117].

##### Cluster 4: Waste and Pollution Dimension

This cluster, colored in yellow, consists of 20 elements. It is focused on pollution and waste management, containing terms such as air quality, disease dispersal and hospital waste management issues. In addition, it refers to the use of new technologies and big data for decision-making to face these problems. Despite the fact that the term “mental health” appears here, it was considered more accurate to move it to the first cluster, associated with the health issues. Cluster 4 is linked to SDGs 8, 9, 11, 12 and 13.

This topic focuses on analyzing the factors and effects of COVID-19 on waste management, based on the expected increase in the occurrence of epidemics [84]. Zambrano-Monserrate et al. [81] predicted the prevalence of negative indirect effects of COVID on the SDGs in the long-term outweighing the possible benefits derived from the shutdown in economic activity. They pointed to the setback in waste management and agreed with Vanapalli et al. [83] on the major problem of plastics´ consumption and hospital waste, which has not been properly treated [118]. In particular, the new technologies and prediction models are being developed to address the sustainability of location-routing problems with COVID waste [119]. This line of research calls for investment in research and development for new personal protective equipment materials that reduce waste generation, with a focus on product lifecycle strategies [120], as well as the use of bio-based solutions to cope with microplastic pollution [121]. In addition, neglected management of this waste has led to unsafe working conditions [122].

Finally, Coccia [82] considered the factors that explain the spread of the virus and pointed to air pollution as a determining element, calling for the prevention strategies in terms of sustainability science and environmental science.

### 3.5. Which Have Been the Main Contributions of the CE to the SDGs during COVID-19?

Fifteen articles specifically investigate the circular economy; most of them consider that the pandemic situation has created a window of opportunity for the transition to the circular model in order to achieve SDGs. This position is most strongly defended by Ibn-Mohammed et al. [49], with the fifth most productive article of the total. The other fourteen articles on the CE receive up to seven times fewer citations, occupying a much smaller space in the research topic. These articles mostly deal with particular case studies in very different sectors. Some significant cases are summarized below.

Rahman et al. [123] studied circularity in Southeast Asian ships that were being dismantled due to the shutdown in the maritime transport of goods. Ducoli et al. [124] investigated the possible use of ashes from sewage sludge contaminated by COVID-19 as a new material for construction. Hoosain et al. [125] analyzed various case studies to show how the technologies of the fourth industrial revolution are allowing the application of circularity principles in a wide variety of sectors and how these technologies are proving to be of key importance in the fight against pandemics. In the same vein, Abdul-Hamid et al. [126] looked into the optimization of palm oil production through digital technologies. Adelodun et al. [97] considered the impact on the agri-food system, highlighting the effectiveness of measures with a CE approach taken in Europe, such as short chains, which are key to achieving the sustainability of the agri-food system. Zanoletti et al. [127] discussed how to ensure the availability of critical raw materials, a growing problem since the pandemic. Girard and Nocca [128] proposed transforming urban planning using a CE approach to improve environmental quality and resilience in the face of future pandemics. Shishkin et al. [129] studied eco-design in air disinfection devices. Sparacino et al. [130] studied the integration of CE in companies, identifying them as key actors in the transition that has accelerated in the wake of the pandemic.

In order to see what circular solutions have been proposed for the challenges posed by COVID-19 to the SDGs, other literature reviews were used. Ten reviews were identified out of the 1093 documents initially detected. Table 8 summarizes the impact that the pandemic has had and in what sense. It also lists the proposed circular solutions and the barriers to making them effective. The main themes were Industry 4.0 technologies, circular models, the effectiveness of the waste hierarchy, the use of new materials and the efficiency of waste management systems.

## 4. Conclusions

This study had two objectives; the first was to explore the impact of COVID-19 on the fulfilment of the SDGs with regard to the research of the scientific community, and the second was to analyze the presence of the CE in the literature. To this end, bibliometric analysis was carried out to answer the following questions.

Q1. What are the main characteristics of this line of research?

Due to the novelty of the phenomenon, the period studied was very short compared to other bibliographical studies, but the number of documents is typical of much more established topics. The 1093 articles studied show a growing and lively trend, as the high number of citations shows.

Q2. What are the main thematic areas and the most relevant publications that address the impact of COVID-19 on the SDGs?

The research of the topic has been characterized by multidisciplinarity, although five areas are deeply involved in its study; the social sciences area is the most productive, followed by environmental sciences, energy, business, management and accounting, medicine, engineering and, finally, economics, econometrics and finance. Thus, the interdependence between the SDGs is clear.

Consequently, although it can be seen that the impact of COVID-19 has been addressed in all the dimensions of sustainability among the 10 most cited articles, it is clear that those linked to the environmental dimension have received greater attention, despite being a virus that has mainly affected the health of the population, which links to the social dimension. This dimension has achieved the least attention, as measured by the total number of total citations.

Q3–Q4. Who are the most productive authors, institutions, countries and journals? Which are the main international cooperation networks?

The majority of the most productive authors are from Asia, the most prolific of which is Dang, T. T., far ahead of other authors such as Leal Filho, W. or Shaw, R. who are in second and third place. However, none of Dang, T. T.’s articles are among the most cited, which is the case with Leal Filho. Cooperation between authors has been very scarce, with only the work of Leal Filho being relevant. The ten most productive journals are Q1, the number one being the generalist journal Sustainability, with eight times more articles than the second most productive journal, the International Journal of Environmental Research and Public Health. In addition, the Chinese Academy of Sciences is the most productive institution and, therefore, China is the most productive country, followed by the United States and the United Kingdom. As in the case of authors, there is a lack of international cooperation between countries and institutions, building weak and fragmented networks.

Q5. What are the main current trends in research on COVID-19 and SDGs?

This question has been difficult to answer, as the same article can cover up to six different SDGs. Scopus has introduced a new feature in its platform for mapping SDGs, but sometimes it assumes that an SDG is being addressed simply because it is mentioned, when, in fact, it receive minimal attention; this can lead to mistakes.

Five clusters were detected that together discuss the three dimensions of sustainability: economic, social and environmental. In general, all articles deal with SDG 3—good health and wellbeing, although SDG 8—decent work and economic growth, SDG 9—industry, innovation and infrastructure and SDG 12—sustainable consumption and production play a major role in the solutions.

Analyzing the SDGs from the perspective of the 5Ps, it is widely accepted that those related to Prosperity, Planet and People have been studied in depth, in that order; meanwhile, the SDGs of Partnership and Peace have received no attention at all. Therefore, it seems that scientific interest has been guided more by the concerns of the market in terms of economic recovery and improving the efficiency of companies under the slogan of sustainability.

With regard to supply chains, the studies analyzed predicted a great opportunity for the promotion of renewable energies, but the current situation of war is making smooth energy transition very difficult and, in some countries has led to the extension of the life of nuclear power plants or to pacts being made with states that violate human rights and reject peace.

Q6. What have been the main contributions of the CE to the SDGs during COVID-19?

As indicated from the outset, SDGs are deeply linked to the CE, which has drawn attention to the CE, especially in the environmental dimension. The CE paradigm and its tools have been part of the solutions to the economic, health and environmental crisis caused by the pandemic. It is seen as the most desirable new model on the basis of the opportunities detected. The articles detected were case studies focused on production changes and the recovery of waste in order to ensure the availability of secondary raw materials and secure supply chains.

This research suggests some limitations that offer potential areas for future lines of research. The main limitations faced by this study are the volatility of the articles (rapid changes in the number of citations and relevance of articles due to the novelty of the topic) and the difficulty in quantifying the number of SDGs addressed and the extent to which the articles are linked to the relevant SDG. For this reason, future lines of research could investigate further the topics detected and analyze the implications for each SDG and the progress being made. This study has analyzed articles only, since the peer review process guarantees a higher scientific quality, as explained in the methodology section. However, the results may be limited due to the fact that excluding other types of documents, e.g., book chapters, may affect the representation of some disciplines such as the humanities. Moreover, the research was restricted to the Scopus database, so including another database such as Google Scholar or Web of Science may significantly change the results. Furthermore, the research included the most common and general terms on the subject, which possibly excludes studies that used more specific terms. Finally, by using this kind of bibliometric analysis, the reduced number of citations interconnecting the various publications may not properly capture the impact of a publication. Indeed, these metrics do not necessarily work well for creative works and may not reflect local cultural practices. In the future, content analysis could be complementary, in order to assess the quality of the research.

Given that the CE has become a new paradigm that advocates the constitution of a new model based on the principles of sustainability, it would be interesting to propose research studies that analyze the impact of COVID-19 on the CE, indicating whether it has contributed to, or, on the contrary, paralyzed the advances that had been taking place up to that point. Neither sustainable finance nor other useful elements to alleviate the economic crisis or to improve the implementation of the CE were found to be relevant in the literature analyzed in this study; thus, future lines of research could address this gap and determine the influence of the pandemic on the concept of the CE.

## Figures and Tables

**Figure 1 ijerph-19-16266-f001:**
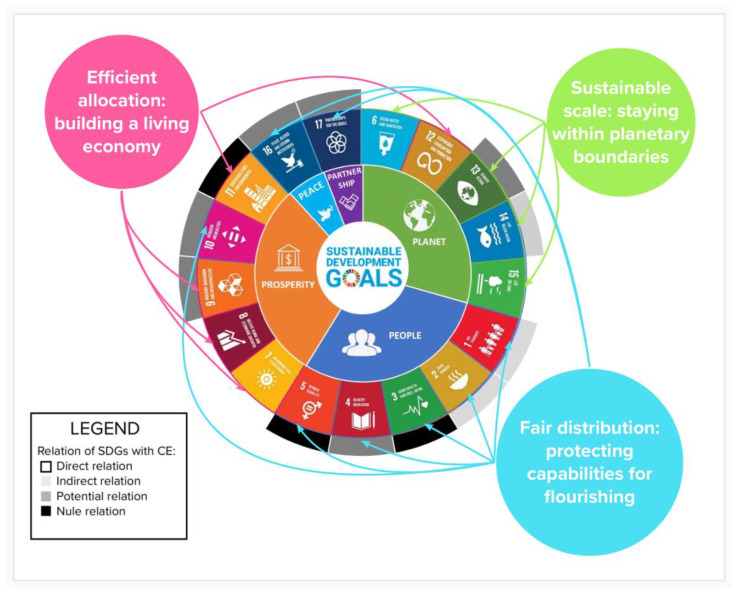
Relation between SDGs and circular economy. Source: our own elaboration adapted on Costanza et al. [7] and Schröder et al. [45].

**Figure 2 ijerph-19-16266-f002:**
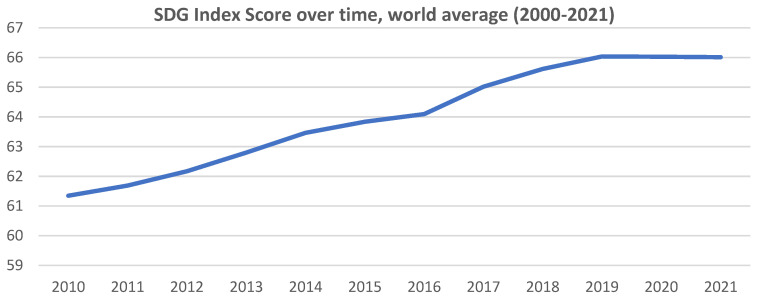
SDGs Index score over time, world average (2010–2021). Source: our own elaboration on Sachs et al. [46].

**Figure 3 ijerph-19-16266-f003:**
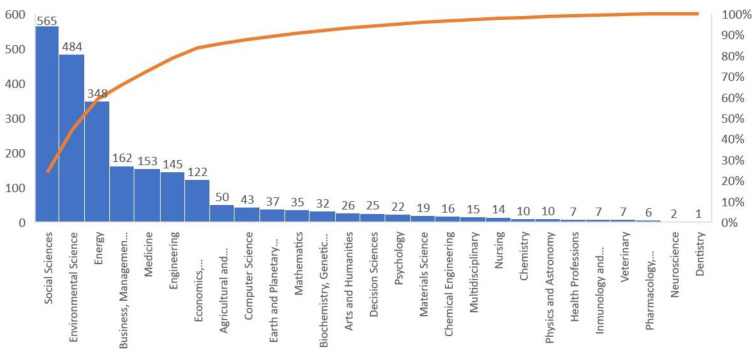
Distribution of articles by subject area.

**Figure 4 ijerph-19-16266-f004:**
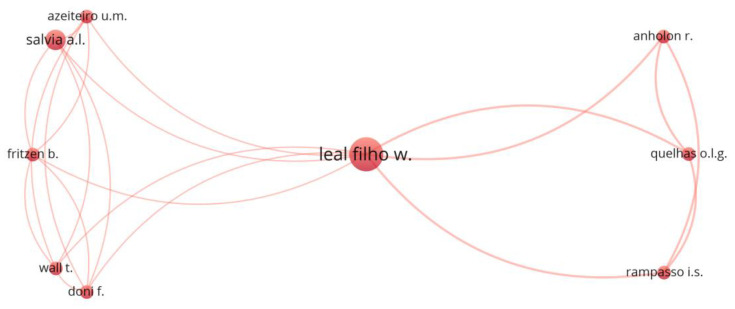
Authors’ cooperation network based on co-authorship.

**Figure 5 ijerph-19-16266-f005:**
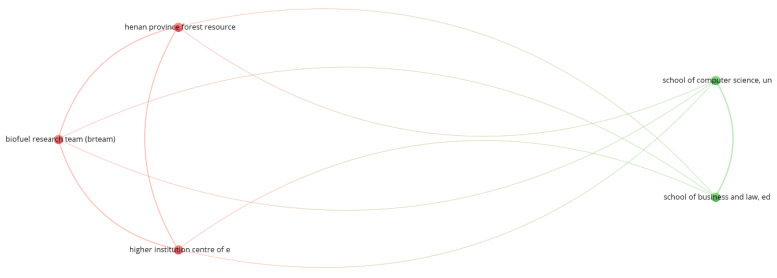
Network of cooperation between institutions based on the co-authorship of articles.

**Figure 6 ijerph-19-16266-f006:**
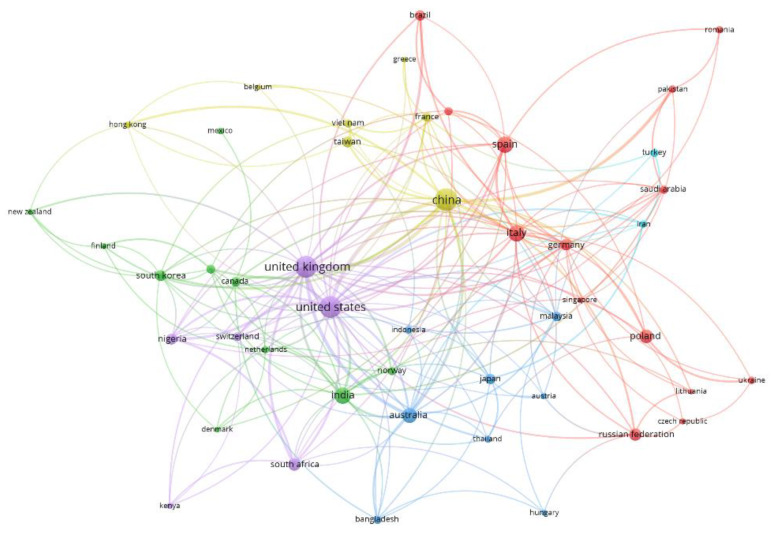
Network of cooperation between countries based on the co-authorship of articles.

**Figure 7 ijerph-19-16266-f007:**
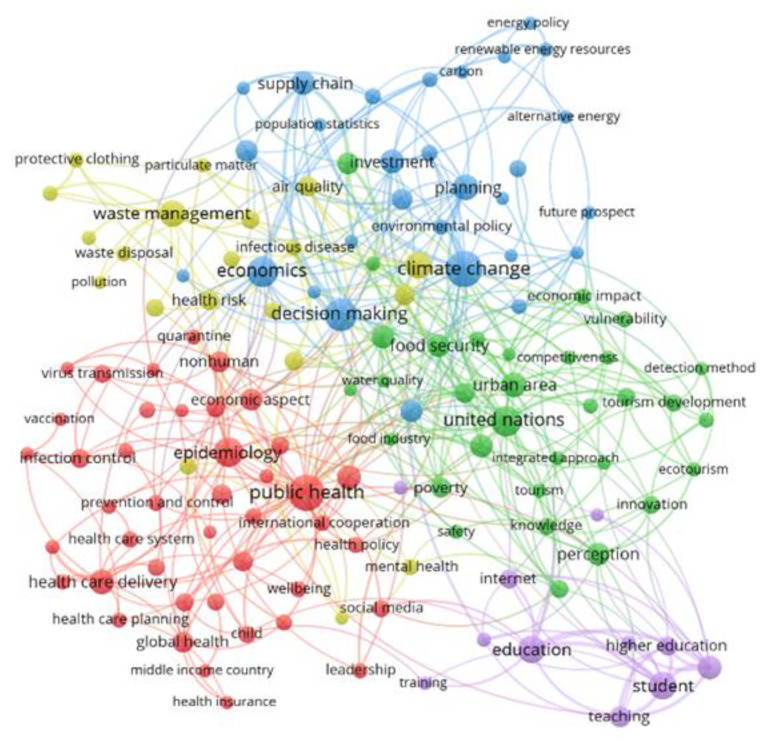
Keywords co-occurrence map.

**Table 1 ijerph-19-16266-t001:** Summary for SPAR-4-SLR protocol application.

Stage	Sub-Stage	
Assembling	Identification	Domain: Topic “Impact of COVID-19 on SDGs”. Research questions: To what extent has the impact of COVID-19 on the SDGs received attention from the research community? Source type: Include only academic sources because they undergo rigorous peer review. Source quality: Scopus, one of the most popular journal quality lists (along with WOS), was selected because it is transdisciplinary and has a broader range of subject areas and categories than WOS, allowing academics to better locate journals according to the areas most relevant to the scope of the review.
	Acquisition	Search mechanism and material acquisition: Scopus database, because it provides bibliometric details to download, which is especially useful for bibliometric analysis. It also avoid results that include predatory journals. Search period: 2020 to 2021. Search keywords: TITLE-ABS-KEY (“COVID-19” OR “coronavirus disease 2019” OR “SARS-CoV-2” OR “coronavirus” OR “coronavirus infection”) AND TITLE-ABS-KEY (“sustainable development goals” OR “sustainable development” OR “sdgs” OR “sdg” OR “Agenda 2030”) (*n* = 2453).
Arranging	Organization	Organizing codes: Data were exported in CSV format, marking citation information, bibliographical information, abstract and keywords, including references.
	Purification	Article type Included: Document type limited to article (*n* = 1483); publication year limited to 2020 and 2021 (*n* = 1148); language limited to English (*n* = 1093); full data.
Assessing	Evaluation	Analysis method: Scientometric; bibliographic modeling was used for co-authorship analysis and topic modeling for cluster analysis and keyword co-occurrence analysis.
	Reporting	Reporting: tables and graphics; VOSViewer software was used for visualization.

Source: our own elaboration according to the SPAR-4-SLR protocol established by Paul et al. [59].

**Table 2 ijerph-19-16266-t002:** General characteristics of the scientific production.

Year	Articles	Authors	Countries	Institutions	Journals	Citations	TC/A	TC/Author
2020	256	844	88	688	159	431	1.68	3.29
2021	837	2812	131	2302	426	4495	5.37	3.36
Total	1093	3574	141	2969	528	4526	4.14	3.27

TC/A: average number of citations per article; TC/Author: average number of citations per author.

**Table 3 ijerph-19-16266-t003:** The 10 most cited articles.

Authors	Year	Title	Citations	Associated SDGs
Zambrano- Monserrate et al. [81]	2020	Indirect effects of COVID-19 on the environment.	545	3, 12, 13
Coccia M. [82]	2020	Factors determining the diffusion of COVID-19 and suggested strategy to prevent future accelerated viral infectivity similar to COVID.	261	3, 9, 11, 14
Vanapalli et al. [83]	2021	Challenges and strategies for effective plastic waste management during and post COVID-19 pandemic.	161	9, 12
Ibn-Mohammed et al. [49]	2021	A critical review of the impacts of COVID-19 on the global economy and ecosystems and opportunities for circular economy strategies.	155	8, 9, 10, 12, 13
Pirouz et al. [84]	2020	Investigating a serious challenge in the sustainable development process: Analysis of confirmed cases of COVID-19 (new type of Coronavirus) through a binary classification using artificial intelligence and regression analysis.	131	3, 8
Ilyas et al. [85]	2020	Disinfection technology and strategies for COVID-19 hospital and bio-medical waste management.	115	3, 12
Yeasmin et al. [86]	2020	Impact of COVID-19 pandemic on the mental health of children in Bangladesh: A cross-sectional study.	91	3
Filho et al. [87]	2020	COVID-19 and the UN sustainable development goals: Threat to solidarity or an opportunity?	91	3
Amankwah- Amoah J. [88]	2020	Stepping up and stepping out of COVID-19: New challenges for environmental sustainability policies in the global airline industry.	87	9, 13
Galvani et al. [89]	2020	COVID-19 is expanding global consciousness and the sustainability of travel and tourism.	79	8, 12

SDGs: Sustainable Development Goals.

**Table 4 ijerph-19-16266-t004:** The ten most productive authors.

Authors	A	TC	TC/A	Institution	C	First A	Last A	H index
Dang, T.T.	5	61	12.20	International University, Vietnam National University, Ho Chi Minh City	Vietnam	2021	2021	6
Leal Filho, W.	5	32	6.40	Hochschule für Angewandte Wissenschaften Hamburg	Germany	2020	2021	3
Shaw, R.	5	26	5.20	Keio University, Graduate School of Media and Governance	Japan	2020	2021	3
Abbas, H.S.M.	4	42	10.50	Huazhong University of Science and Technology	China	2021	2021	3
Nguyen, N.-A.-T.	4	55	13.75	National Kaohsiung University of Science and Technology	Taiwan	2021	2021	4
Nhamo, G.	4	98	24.50	University of South Africa, Institute for Corporate Citizenship	South Africa	2021	2021	2
Wang, C.-N.	4	55	13.75	National Kaohsiung University of Science and Technology	Taiwan	2021	2021	4
Adelodun, B.	3	81	27.00	Kyungpook National University	South Korea	2020	2021	3
Ali, S.M.	3	104	34.67	Bangladesh University of Engineering and Technology	Bangladesh	2021	2021	3
Allam, Z.	3	94	31.33	Deakin University	Australia	2021	2021	2

(A): number of research articles published; (TC): total number of citations; (TC/A): average number of citations per article; (C): country; (First A): first article published; (Last A): last article published; (H index): Hirsch index, which represents the weight of an author in the line of research.

**Table 5 ijerph-19-16266-t005:** The ten most productive institutions.

							TC/A	
Institution	C	A	TC	TC/A	H index	CI (%)	CI	NCI
Chinese Academy of Sciences	China	15	167	11.13	7	60.0%	14.00	6.83
Organisation Mondiale de la Santé	Switzerland	9	16	1.78	2	88.9%	1.88	1.00
University of Sussex	United Kingdom	8	74	9.25	6	75.0%	8.67	11.00
University of Pretoria	South Africa	8	28	3.50	3	25.0%	8.00	2.00
University College London	United Kingdom	8	38	4.75	4	87.5%	4.57	6.00
UNSW Sydney	Australia	8	120	15.00	3	87.5%	16.71	3.00
London School of Hygiene & Tropical Medicine	United Kingdom	8	44	5.50	2	87.5%	6.00	2.00
University of South Africa	South Africa	7	75	10.71	3	71.4%	15.00	0.00
Texas A&M University	United States	7	103	14.71	4	100.0%	14.71	0.00
Russian Academy of Sciences	Russian Federation	7	28	4.00	3	42.9%	8.00	1.00

(C): country; (A): total number of published articles; (TC): total number of citations; (TC/A): average number of citations per published article; (H index): H index in the line of research; (CI): cooperation index; (TC/A CI): average number of citations of articles in international cooperation; (TC/A NIC): Average number of citations of articles without international cooperation.

**Table 6 ijerph-19-16266-t006:** The ten most productive countries.

Country	A	TC	TC/A	H Index	NC	Main Collaborators	CI (%)	TC/A	
								CI	NCI
China	147	1245	8.47	21	45	United States, United Kingdom, Australia, Pakistan, Taiwan	53.06%	11.44	5.12
United States	139	1388	9.99	19	66	United Kingdom, China, India, Switzerland, Australia	67.63%	11.50	6.82
United Kingdom	135	1252	9.27	19	70	United States, Australia, China, Germany, Nigeria	74.07%	9.61	8.31
Spain	80	524	6.55	12	43	United Kingdom, Italy, United States, France, Netherlands	41.25%	9.30	4.62
Italy	78	995	12.76	15	39	United States, Spain, United Kingdom, Portugal, Australia	42.31%	17.09	9.58
India	76	675	8.88	15	47	United States, United Kingdom, Australia, China, Switzerland	47.37%	10.50	7.43
Australia	64	593	9.27	13	59	United Kingdom, China, United States, India, Bangladesh	73.44%	11.23	3.82
Poland	53	386	7.28	10	32	Italy, Ukraine, United States, China, Estonia	33.96%	13.00	4.34
Russian Federation	44	94	2.14	6	16	China, United Kingdom, Austria, Czech Republic, Italy	38.64%	3.82	1.07
South Africa	44	379	8.61	11	41	United Kingdom, United States, Nigeria, Australia, Germany	54.55%	8.79	8.40

(C): country; (A): total number of published articles; (TC): total number of citations; (TC/A): average number of citations per published article; (H index): H index in the line of research; (NC): total number of international collaborators; (CI): cooperation index; (TC/A CI): average number of citations of articles in international cooperation; (TC/A NIC): average number of citations of articles without international cooperation.

**Table 7 ijerph-19-16266-t007:** The ten most productive journals.

Journal	A	TC	TC/A	H Index Articles	H Index Journal	SJR	C
*Sustainability Switzerland*	218	1241	5.69	20	109	0.66 (Q1)	Switzerland
*International Journal Of Environmental Research And Public Health*	25	234	9.36	8	138	0.81 (Q1)	Switzerland
*Energies*	22	108	4.91	6	111	0.65 (Q1)	Switzerland
*Journal Of Cleaner Production*	19	434	22.84	9	232	1.92 (Q1)	United Kingdom
*Sustainable Cities And Society*	19	276	14.53	11	82	2.02 (Q1)	Netherlands
*Science Of The Total Environment*	17	1339	78.76	12	275	1.81 (Q1)	Netherlands
*Environmental Research*	9	137	15.22	7	149	1.51 (Q1)	United States
*Sustainable Production And Consumption*	9	118	13.11	4	38	1.36 (Q1)	Netherlands
*Frontiers In Psychology*	8	23	2.875	3	133	0.87 (Q1)	Switzerland
*Renewable And Sustainable Energy Reviews*	8	90	11.25	6	337	3.68 (Q1)	United Kingdom

(A): total number of published articles; (TC): total number of citations; (TC/A): average number of citations per published article; (H index articles): H-index of the articles of the line of research; (H index journal): H-index of the journal; (SJR): SCImago Journal Rank.

**Table 8 ijerph-19-16266-t008:** The main contributions of the CE to the SDGs during COVID-19.

Article	Goal	D	Objective	COVID-19	CE
Ibn-Mohammed T. et al. [49]	8; 9; 12	EC-EN-SO	To examine the interplay of literature on public health, socio-economic and environmental dimensions of COVID-19 impacts with CE approaches and to determine if the recovery should be targeted towards constructing a more resilient low-carbon CE.	Weakening of the global value chain. Key sectors affected: aviation, tourism and health care.	Climate change mitigation and adaptation, economic resilience, social inclusion, local development.Actions: material recirculation, material efficiency of products, circular business models, rethinking the optimal size of circles.A need for innovations to address challenges in plastic waste collection, segregation and treatment in the existing waste management system. Investments in circular technologies such as feedstock recycling, improving the infrastructure and environmental viability of existing techniques. Transition towards environmentally friendly materials such as bioplastics.
				Change in consumption and production patterns; temporary reduction of air and noise pollution.	Technical implementation, behavioral change, financial and intellectual investments, policy and regulations, market dynamics, socio-cultural considerations, operational cost of transforming from the linear economy.
Vanapalli K.R., et al. [83]	8; 9; 12	EN-EC	To highlight the implications of COVID-19 on plastic waste generation and the waste management systems.	Mismanagement of plastic waste threatens the environment. The fear of transmission has shifted our behavioral patterns: increase in food packaging waste and single-use plastic bags, personal protective equipment, medical packaging waste, change to throw away culture, panic buying.	A need for innovations to address challenges in plastic waste collection, segregation and treatment into the existing waste management system. Investments in circular technologies such as feedstock recycling, improving the infrastructure and environmental viability of existing techniques. Transition towards environmentally friendly materials such as bioplastics.
					Technological: single use plastic and multi-layered plastics have a low reward-to-effort ratio in their collection, high preprocessing costs, technological constraints and weak integral structure, in addition to the decrease in cost of virgin plastic and the fear of viral transmission during collection.
Hoosain M.S., et al. [125]	8; 9; 12	EC	To understand how 4.0 technological advancements and innovative techniques are used in different sectors to provide an opportunity to resolve the challenges of the SDGs.	Increased energy consumption and emissions in data centers, due to the rise of cloud computing, online meetings, databases and IoT systems.	EC offers tools for Industry 4.0 to remedy its energy consumption. Companies that invested in 4.0 tech were more resilient during the pandemic. They can use digital, physical or biological technologies to improve their economic and environmental benefits with circular thinking.
				Huge advances in sustainability thanks to 4.0 technologies applied to address some of the effects of COVID-19.	Points to work on: government interventions; policies in the form of cross-departmental collaborations and incentives towards businesses; economic, social and environmental impacts on local communities need to be further assessed; education; tech companies require and need to collect more data; more talent is required to improve existing technologies.
Adelodun B., et al. [97]	2; 8; 9; 12	EC-EN	To explore policy framework and select feasible actions that are being adopted during the COVID-19 pandemic, which could potentially reduce emissions even after the pandemic to promote a resilient and sustainable agri-food system.	Some implemented measures to combat the spread of COVID-19 disrupted agricultural activities and the food supply chain. The pandemic has highlighted food insecurity. Determinants: rapid urban development, overpopulation, huge global energy consumption, dense settlements, natural resource depletion and GHG emissions.	Opportunity for local food production, inclusion of Agriculture 4.0 tools (i.e., precision farming, remote sensing, vertical farming), application of AI to avoid food waste and achieve an efficient agri-food supply chain.
				The reduction in ecosystem carbonization. Industrial and household food waste generations were greatly reduced and lifestyles have been altered due to the lockdown.	Stakeholders’ willingness to cooperate, investment and available technologies.They must join forces to apply 4.0 technologies to the agri-food system value chain and ensure food security while reducing pollution of activities.
Hassan A., et al. [131]	8	EC	To explore the role of non-financial business reporting as responsible for crisis such as COVID-19.	Economic shutdown.	CE provides the tools to adopt more sustainable economic models in companies.
				It has evidenced the relevance of non-financial activities and corporate social responsibility. By paying attention to biodiversity and ecosystem health, companies can recognize the risks and opportunities, anticipate new markets and mitigate their impacts.	It is necessary to improve the measure of circularity. Companies should adopt the CE concept for sustainable business models and report on biodiversity and extinction accounting in more structured and mandatory ways by producing integrated reports to create value in the short, medium and long terms.
Zanoletti A., et al. [127]	8; 9; 12; 13	EC-EN	To present the impact of the pandemic on the supply of critical raw materials and to propose some actions that should be pursued in a post-pandemic renaissance scenario to increase raw materials availability.	Disruption of global supply chains for critical raw materials, especially those sourced from third countries. In a post-pandemic scenario, demand for critical raw materials is expected to increase rapidly. The global impact of the pandemic on mining projects worldwide was estimated at over EUR 7 billion.	The EC provides the necessary tools to secure the supply of critical materials, while contributing to climate change mitigation through the recirculation of materials.
					Deficient waste recycle systems are unable to recover critical raw materials, especially in the case of electronic waste. The mining sector is energy intensive. It is necessary to improve waste mining.
Barone A.S., et al. [132]	2; 8; 9; 12	EC-EN	To explore opportunities on active green-based packaging beyond the COVID-19 pandemic, applications in food, and perspectives in the circular economy.	Increased use of packaging.	Circular business model: introducing biobased packaging in replace of plastic, reducing the waste.
				It has made the population more aware of the relevant role of packaging for protection and conservation of food.	Technological gaps and the high costs associated with alternative natural materials, particularly regarding difficulties related to production on an industrial scale, reduced barrier properties and the guarantee of the stability of the bioactive compounds on active packages.
Girard L.F. & Nocca F. [128]	8; 11	EC-EN	To understand the impact of climate change and health in urban areas and to propose a new evaluation tool for governance.	The health emergency has placed the human dimension at the center of development strategies for cities.	New urban development models based on circular models.
				Evidenced the necessity of human-scale projects in cities.	Difficulties in translating the evidence relating to the link between the CE and health issues in quantitative or monetary terms.Hybrid evaluation tools are needed to capture the multidimensional impacts and related links of the implementation of the circular model.
Roque A.J., et al. [133]	6; 7; 8; 9; 11; 12	EC-EN	To present an overview of sustainable technologies and management practices regarding the reuse of several types of waste in geo-environmental projects.	It has accentuated some environmental problems through the increase of waste production, significant reduction in waste recycling and entry of disinfectants into soils and waters.	Opportunity for environmental geotechnics in cities and building sector: re-entry of construction and demolition of wastes, excavated materials, industrial wastes and marine sediments into the production cycle and the reuse of existing foundations; landfill mining.
				Reduction of air pollution and environmental noise during lockdown; cleaner beaches and coastal waters as a result of reduced tourism.	In foundation reuse design: the lack of credible information on the original design and construction of the existing foundations. In general: consumption patterns, landfill availability. Remove administrative discrepancies across European countries for sustainable practices.
Pérez-Peña M.C., et al. [134]	1; 8; 10; 12; 13	SO-EC- EN	To present the current state of scientific research related to inequality, poverty and climate change, and propose lines of improvement that can contribute to the achievement of SDGs 1, 10 and 13.	It highlights the deep inequalities affecting our economies, health and quality of life. It spread extreme poverty in overcrowded cities, especially affecting people dependent on informal jobs. The groups most affected were children, women, the elderly, migrants and people with health problems. Problems: contaminated water, food insecurity, malnutrition and hunger, growing economic inequality, especially in countries dependent on agriculture as their main activity.	EC strategies based on climate change mitigation, more sustainable production and reduction of negative externalities, while creating employment opportunities, can address the growing disadvantages for equity, health and food security.

D: dimensions of sustainability addressed; COVID-19: in red, the negative effects; in green, the positive ones; EC: in green, the opportunities; in red, the barriers.

## Data Availability

The raw data supporting the conclusions of this article will be made available by the authors, without undue reservation.

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
