# Peer review of "The Impact of COVID-19 on the Sustainable Development Goals: Achievements and Expectations"

_ijerph, 2022, doi:10.3390/ijerph192316266_

Round 1
Reviewer 1 Report
Thank you for giving me the opportunity to review the manuscript titled “The impact of COVID-19 on the Sustainable Development Goals: Achievements and Expectations”. The manuscript is well written and provides important bibliometric information about SDGs research conducted during Covid 19 time period.
Following are the few issues which need attention
1- Percentages presented in figure 3 are confusing.
2- What is meant by “ODS” in table no 3? Provide that in the table footer.
3- Based on the table no 4 on page no 11 following information is not correct “Dang, T. T. standing out by far, with twice as many publications as the rest, despite being published in the last year.”
Shaw R. from Japan has also published 5 papers.
4- You need to clearly list the limitation of this investigation which might include
a. your search strategy as there is no one article database that includes all articles published and which can analyze all the relations between them.
b. In some disciplines, particularly in the humanities, investigations rely more heavily on particular formats for scholarly output, especially books and book chapters.
c. Publications in some disciplines do not rely heavily on citations to other work. Thus, the reduced number of citations interconnecting publications may not properly capture the impact of a publication. Moreover, these metrics do not necessarily work well for creative works and may not reflect local cultural practices.
5- There are some specific keywords identified in the Scopus database for mapping each SDG. Have you considered those SDG-specific keywords for mapping the literature?
Author Response
Dear reviewer,
Thank you for taking the time to read our manuscript carefully. We appreciate your comments, which will certainly improve the quality of the document. The changes made in the manuscript have been included using the Track changes functionality of Word. Below we respond to your comments one by one:
1. Percentages presented in figure 3 are confusing.
We have corrected the dimension of figure 3 so that the numbers above the bars are visible. They represent the number of items per subject area.
2. What is meant by “ODS” in table no 3? Provide that in the table footer.
The mistake with SDG has been corrected. For further clarification the abbreviation now appears in the abstract and in the introduction.
3. Based on the table no 4 on page no 11 following information is not correct “Dang, T. T. standing out by far, with twice as many publications as the rest, despite being published in the last year.” Shaw R. from Japan has also published 5 papers.
The sentence in lines 336-339 has being replaced by “Among the ten most productive authors, seven come from the Asia and one of these, Dang, T. T., leads the list ,with twice as many citations as the second and third most productive authors, despite having only been published in the last year.”
4. You need to clearly list the limitation of this investigation which might include
- your search strategy as there is no one article database that includes all articles published and which can analyze all the relations between them.
- In some disciplines, particularly in the humanities, investigations rely more heavily on particular formats for scholarly output, especially books and book chapters.
- Publications in some disciplines do not rely heavily on citations to other work. Thus, the reduced number of citations interconnecting publications may not properly capture the impact of a publication. Moreover, these metrics do not necessarily work well for creative works and may not reflect local cultural practices.
Thank you for your suggestions. We have included more limitations in lines 664-676:
“This study has analysed articles only, since the peer review process guarantees a higher scientific quality, as explained in the methodology section. However, the results may be limited due to the fact that excluding other types of documents, e.g. book chapters, may affect the representation of some disciplines such as the humanities. Moreover, the research was restricted to the Scopus database, so including another database such as Google Scholar or Web of Science may significantly change the results. Furthermore, the research included the most common and general terms on the subject, which possibly excludes studies that have used more specific terms. Finally, by using this kind of bibliometric analysis, the reduced number of citations interconnecting the various publications may not properly capture the impact of a publication. Indeed, these metrics do not necessarily work well for creative works and may not reflect local cultural practices. In the future, content analysis could be complementary, in order to assess the quality of the research”.
5. There are some specific keywords identified in the Scopus database for mapping each SDG. Have you considered those SDG-specific keywords for mapping the literature?
Thank you for raising this relevant point. In this research we have tried to include all possible variations of the SDG term as is shown in lines 179-180 “("sustainable development goals" OR "sustainable development" OR "sdgs" OR "sdg" OR "Agenda 2030")”. Using terms specific to each SDG would be too broad and could lead to errors in the database by including issues that are inappropriate to our object of study. Consequently, we have proposed this as future lines of research and limitations of the study.
Reviewer 2 Report
Dear Authors,
I am sorry to say that the article submitted to me for review, in my opinion does not meet the requirements of a scientific article. It is primarily a compilation of publications dedicated to a selected topic, without a specific scientific goal and it does not bring any cognitive value in this topic. As indicated, "This paper shows the development of scientific activity since the beginning of the pandemic and how the effect of the pandemic on the achievement of the SDGs has been addressed."
The article does not contain any of the authors' own research and conclusions, which constitute the scientific value of the article and at the same time contribute to the development of the scientific field. The conducted review may be considered as part of the article, but not as an independent scientific study ("this research applies bibliometric analysis and the literature review").
I am far from the opinion that each scientific article must contain strictly quantitative research, but it must provide much more than an answer to questions such as: "Who are the most productive authors, institutions, countries and journals? Which are the main international cooperation networks? ". By the way, for the first time I come across the research on the productivity of "authors, institutions, countries and journals".
The discussed topic is interesting and up-to-date, I encourage the Authors to develop it and conduct their own research in this area.
Author Response
Dear reviewer,
Thank you for taking the time to read this article. Our research is in fact a review in which we have applied scientometrics. This methodology mainly uses quantitative data and it is a well-established research typology as shown in the literature that we have used. We regret that you do not consider the manuscript suitable, though we are grateful for your comments. Notwithstanding, we have incorporated some limitations in relation to the scientometric method used in lines 664-676:
“This study has analysed articles only, since the peer review process guarantees a higher scientific quality, as explained in the methodology section. However, the results may be limited due to the fact that excluding other types of documents, e.g. book chapters, may affect the representation of some disciplines such as the humanities. Moreover, the research was restricted to the Scopus database, so including another database such as Google Scholar or Web of Science, may significantly change the results. Furthermore, the research included the most common and general terms on the subject, which possibly excludes studies that have used more specific terms. Finally, by using this kind of bibliometric analysis, the reduced number of citations interconnecting the various publications may not properly capture the impact of a publication. Indeed, these metrics do not necessarily work well for creative works and may not reflect local cultural practices. In the future, content analysis could be complementary, in order to assess the quality of the research”.
Reviewer 3 Report
Dear Authors,
Please find below and attached my comments and suggestions for your work.
Good luck!
Kind regards,
The Reviewer
Review Report Form
Journal: IJERPH (ISSN 1660-4601)
Manuscript ID: ijerph-2000603
Type: Article
Title: The impact of COVID-19 on the Sustainable Development Goals: Achievements and Expectations
Authors: Cathaysa Martín-Blanco * , Montserrat Zamorano , Carmen Lizárraga , Valentin Molina
Section: Global Health
Special Issue: Vulnerable Communities and Public Health
Submission Date: 19 September 2022
Dear Authors,
I have carefully analyzed your article entitled “The impact of COVID-19 on the Sustainable Development Goals: Achievements and Expectations”.
Congratulations for your work and valuable insights reflected in the content of the manuscript!
The structure of my Review Report Form takes into consideration two sections, namely: (A.) General overview of the article and strong points; and (B) Suggestions meant to improve your current manuscript.
(A.) General overview of the article and strong points:
Ø General background of the study: According to the authors, the COVID-19 pandemic has a significant impact on almost all the SDGs, leaving no country indifferent. In continuation, the authors mentioned that it caused a shift in political agendas, but also in lines of research. What is more, the authors stressed their belief according to which at the same time, the world is trying to transition to a more sustainable economic model.
Ø Aim of the study: According to the authors’ notes, this paper shows the development of scientific activity since the beginning of the pandemic and how the effect of the pandemic on the achievement of the SDGs has been addressed. Also, the role of the Circular Economy paradigm is discussed.
Ø Methodology: The authors created a systematic literature review and bibliometric analysis was conducted to determine the lines that have been addressed, as well as neglected targets, using VOSviewer for data visualisation.
Ø Results of the study: In terms of the findings, the authors have mentioned the following ones: Five clusters have being detected and grouped according to the triple dimension of the sustainability. Also, the extent of the effects of the health, economic and social crisis resulting from the pandemic, in addition to the climate crisis, is still uncertain, but it seems clear that the main problems are inefficient waste management, supply chain problems, adaptation to online education and energy problems.
(B) Suggestions meant to improve your current manuscript:
Distinguished Authors I would kindly like to suggest the following aspects:
(1.) Closely analyzing the article, since there are some English language improvements and slight corrections that need to be taken care of. Thus, my recommendation would be to carefully proofread the entire manuscript.
(2.) Also, I have closely analyzed the format of the article, in order to check whether it follows the guidelines which are specific to the publisher. Thus, I have noticed that the current form of your work needs improvement in this regard. So, my kind suggestion is to closely analyze again the guidelines belonging to the publisher, since the article should fit exactly the publisher’s guidelines. For instance, the keywords, the subsections, the references, currently do not fit the style and the requirements of the publisher. Also, it would be highly recommendable to include in the abstract of your study more highly relevant details that refer to the research objectives and the methodology used. This would definitely be considered a plus for your scientific work.
(3.) In continuation, the suggestion would also be inserting in your article a few ideas concerning the correlation between effects of the COVID-19 pandemic and the COVID-19 global crisis, sustainability and sustainability assessment, Sustainable Development Goals, while focusing on the research on the influence mechanism of digital finance on the high-quality development of state-owned enterprises, the perspective of financing constraints, since these are key focuses these days. In this context, I had the chance to read a few interesting scientific works recently, among which I would like to mention: Measuring Progress Towards the Sustainable Development Goals: Creativity, Intellectual Capital, and Innovation. In C. Popescu (Eds.), Handbook of Research on Novel Practices and Current Successes in Achieving the Sustainable Development Goals (pp. 125-136). IGI Global. https://doi.org/10.4018/978-1-7998-8426-2.ch006; OECD. Measuring the Impacts of Business on Well-Being and Sustainability. https://www.oecd.org/statistics/Measuring-impacts-of-business-on-well-being.pdf; OECD. 2022. Toward sustainable economic development through promoting and enabling responsible business conduct. https://www.oecd-ilibrary.org/sites/f7813858-en/index.html?itemId=/content/component/f7813858-en.
Dear Authors, congratulations once again for your work and valuable insights reflected in the content of the manuscript, and I hope my comments will be of value to you!
Kind regards,
The Reviewer
Author Response
Dear Reviewer,
Thank you for your valuable suggestions and for taking the time to make substantial contributions that will certainly improve the quality of our manuscript. The changes made in the manuscript have been included using the Track changes functionality of Word. Below we respond to your comments one by one:
1. Thank you for pointing out this. The entire manuscript has been carefully proofread.
2. The instructions to the author stated that "IJERPH now accepts free-format submissions". This caused us some confusion. Observing your suggestions, the guidelines have been analysed, Vancouver Style has been used and now, all keywords, subsections and references fit the style and the requirements of the publisher.
3. Thank you for sharing with us these interesting and relevant scientific papers in a very rich line of research. We have included the scientific works that you brought to us in lines 68-71:
“This shift is related to efforts to embed sustainable finance in both private and public organisations, as well as policy initiatives to encourage responsible business conduct for sustainable development [16–18]”
and also in the proposals for future lines of research in lines 681-684:
“Neither sustainable finance nor other useful elements to alleviate the economic crisis or to improve the implementation of the CE have been found to be relevant in the lit-erature analysed in this study, thus future lines of research could address this gap and determine the influence of the pandemic on the concept of the CE”.
Reviewer 4 Report
General assessment: Analysis is interesting. Survey of literature is comprehensive and well structured. Visualisation of several results is well prepared. Research questions are answered.
General remarks:
1) There should be an empty line between table and the next paragraph, not between the title and sources,
2) References should be made in numbers [in brackets], according to the template of the journal
3) More thorough editing for English content is suggested.
Minor (editing) remarks (it is not a compete list!!!) can be found in the attachement.

Author Response
Dear reviewer,
Thank you for carefully reading our manuscript. We are grateful for your valuable comments, which we have taken into consideration. The changes made in the manuscript have been included using the Track changes functionality of Word. Below we address them one by one:
1. There should be an empty line between table and the next paragraph, not between the title and sources
The errors in the tables have been amended as indicated, eliminating the empty line where it was unnecessary and introducing it where it was.
2. References should be made in numbers [in brackets], according to the template of the journal
We have modified the references, using a numbered referencing style (Vancouver).
3. More thorough editing for English content is suggested.
Thank you for highlighting this. The entire manuscript has been carefully proofread.
Round 2
Reviewer 2 Report
Dear Authors,
I have not received a response to my comments, and the key concerns I raised have not been clarified. What is the scientific value of your article? Who will quote it and for what purpose? To indicate that the most "productive" author is Mr./Ms. X from country Z? That Mr. Y mainly works with Mrs. Z? This type of analysis may be useful in the field of library science. Research comparing other people's publications usually refers to the results of research by other authors - here hypotheses are formulated and research grouped confirming or not their truth. If the most frequently undertaken problems and the most frequently obtained results were compared, the bibliometric analysis would indeed be justified.
By the way - the concept of "productivity of authors, countries, .." etc., which is a key concept in the article, has not been explained. The text that was added to the article does not relate to the comments in my review.
The color schemes linking the most productive countries and authors are very nicely done and certainly required a lot of work. However, I am sorry to say that I still do not know what they ultimately serve and what results from them.
Best regards,
Reviewer
Author Response
Dear Reviewer,
We sincerely appreciate the time and effort you have dedicated to providing feedback on our manuscript and are grateful for the insightful comments on and valuable improvements to our paper. Please see below, in blue, a point-by-point response to your comments and concerns. All line numbers refer to the re-revised manuscript file.
- What is the scientific value of your article?
The scientific value of our article has been specified in lines 143-153:
“In general, studies that apply a systematic review of the literature are valuable to understand the leading edge research in a field, but an additional analysis of the liter-ature using bibliometric methods can provide additional results that have not been detected in the other reviews [53]. Thus, bibliometric research, analyzing large vol-umes of scientific data, has an increasing scientific value in responding to the Public Health Emergency of International Concern [54]. In this case, the pandemic has gener-ated enormous attention in the scientific community, reflected in the large volume of articles published during the last two years, but it has been shown that COVID-19-related reviews have been limited and fragmented in particular areas [12]. This paper fills this gap, applying bibliometric analysis, to explore the implication of the impact of COVID-19 on the fulfillment of the SDGs and the presence of the Circu-lar Economy (CE) in that literature.”
- Who will quote it and for what purpose? To indicate that the most "productive" author is Mr./Ms. X from country Z? That Mr. Y mainly works with Mrs. Z? This type of analysis may be useful in the field of library science.
Bibliometric analysis has become an essential tool for assessing and analyzing researchers’ production, collaboration between institutions, and the impact of state scientific investment in national R&D productivity or academic quality (Moral-Muñoz et al., 2020). We consider that other researchers interested in monitoring the trends and changes that may occur in this line of research will quote the paper. In addition, it may be of interest to identify the most relevant authors, institutions, networks and countries in the research, or those that do not have a presence or whose contribution is still low. Moreover, those researchers analyzing policy formulation in response to sustainability challenges can quote this analysis.
Detecting the most productive authors is relevant in order to identify who are the authorities in a particular field and to gauge how the development of the scientific production is progressing. Therefore, productivity is a key concept in bibliometric analysis that makes use of indicators such as the number of citations or the h index.
Moral-Muñoz, J. A., Herrera-Viedma, E., Santisteban-Espejo, A., & Cobo, M. J. (2020). Software tools for conducting bibliometric analysis in science: An up-to-date review. Profesional de La Información, 29(1), Article 1. https://doi.org/10.3145/epi.2020.ene.03
- Research comparing other people's publications usually refers to the results of research by other authors - here hypotheses are formulated and research grouped confirming or not their truth. If the most frequently undertaken problems and the most frequently obtained results were compared, the bibliometric analysis would indeed be justified.
Section 3.4. shows the results on the issues that have attracted the most interest from the three dimensions of sustainability and illustrates some of the main solutions that have been found. To clarifying this issue, we have changed the section name (lines 426-427):
“3.4. Which have been the most frequently undertaken problems and results? Which SDGs have received the most attention?”
In lines 441-443, we have specified that “This section contains the most frequently undertaken problems and the most frequently obtained results by cluster”.
Moreover, we have improved the section on Circular Economy. We have inserted a table that summarizes the main articles that study the role of CE in the research concerning the impact of COVID-19 on the fulfillment of SDGs. For each article, the table lists the problems caused by COVID which Goal has been impacted and in what sense, and the role of CE. This is table 8 on line 608 and an explanatory paragraph on lines 601 to 607.
- By the way - the concept of "productivity of authors, countries, .." etc., which is a key concept in the article, has not been explained.
The indicators used to measure productivity are particular to each case and they are explained at the bottom of each table. For example, in the case of most productive authors, Table 4 in line 329, we use four indicators: the number of publications, the total number of citations, the average number of citations per publication and the H index of the author. All of these are measured in relation to the line of research.
Best regards.